# Training deep segmentation networks on texture-encoded input: application to neuroimaging of the developing neonatal brain

**Ahmed E. Fetit**                   A.FETIT@IMPERIAL.AC.UK
**John Cupitt**                     J.CUPITT@IMPERIAL.AC.UK
**Turkay Kart**                     T.KART@IMPERIAL.AC.UK
**Daniel Rueckert**                 D.RUECKERT@IMPERIAL.AC.UK

*Biomedical Image Analysis Group, Department of Computing, Imperial College London, London, SW7 2AZ, United Kingdom.*

## Abstract

Standard practice for using convolutional neural networks (CNNs) in semantic segmentation tasks assumes that the image intensities are directly used for training and inference. In natural images this is performed using RGB pixel intensities, whereas in medical imaging, e.g. magnetic resonance imaging (MRI), gray level pixel intensities are typically used. In this work, we explore the idea of encoding the image data as local binary textural maps prior to the feeding them to CNNs, and show that accurate segmentation models can be developed using such maps alone, without learning any representations from the images themselves. This questions common consensus that CNNs recognize objects from images by learning increasingly complex representations of shape, and suggests a more important role to image texture, in line with recent findings on natural images. We illustrate this for the first time on neuroimaging data of the developing neonatal brain in a tissue segmentation task, by analyzing large, publicly available T2-weighted MRI scans (n=558, range of postmenstrual ages at scan: 24.3 - 42.2 weeks) obtained retrospectively from the *Developing Human Connectome Project* cohort. Rapid changes in visual characteristics that take place during early brain development make it important to establish a clear understanding of the role of visual texture when training CNN models on neuroimaging data of the neonatal brain; this yet remains a largely understudied but important area of research. From a deep learning perspective, the results suggest that CNNs could simply be capable of learning representations from structured spatial information, and may not necessarily require conventional images as input.

**Keywords:** Segmentation, convolutional neural networks, local binary patterns, texture, neuroimaging, neonatal, developing brain.

## 1. Introduction

One widely accepted explanation of the effectiveness of convolutional neural networks (CNNs) in classification and semantic segmentation tasks is the so-called *shape hypothesis* (Geirhos et al., 2019); low-level shape features are combined in increasingly complex hierarchies until the object can be readily classified or detected (LeCun et al., 2015). Whilst this hypoth-

esis is supported by a number of empirical findings (Zeiler and Fergus, 2014; Ritter et al., 2017), recent work in the machine learning literature suggests an important role for visual *texture* in object recognition tasks. For instance, Brendel and Bethge showed that CNNs can achieve high classification accuracy on the publicly available ImageNet (Russakovsky et al., 2015) data in settings where the model is effectively constrained to recognizing local textural patches (Brendel and Bethge, 2019). Recent analysis by Geirhos and colleagues also supports this claim, and illustrates that ImageNet-trained CNNs are strongly biased towards the recognition of textural representations as opposed to shapes; an observation termed *texture hypothesis* by the authors (Geirhos et al., 2019). A preliminary study by Pawlowski and Glocker also supports the texture hypothesis in the context of medical imaging, and shows that textural information alone could indeed be sufficient for regression and classification tasks when using T1-weighted brain MRI (Pawlowski and Glocker, 2019).

We contribute to the ongoing debate on the role of texture in deep learning within the context of neuroimaging of the developing brain. This is an important medical application; visual characteristics of neonatal and fetal brains are significantly different from those of adult brains in terms of size, morphology, and white/gray matter intensities (Makropoulos et al., 2018) *à la* visual shape and texture. Importantly, changes in visual characteristics occur rapidly during brain development as a result of the continuous decrease of water content within the brain and the process of myelination (Serag, 2013); Figure 1.

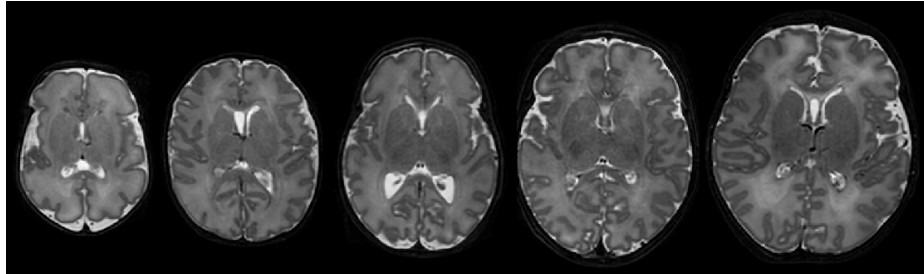

Figure 1: Axial slices from T2-weighted brain MRI scans of neonates, postmenstrual ages at scan: 32, 34, 45, 38, and 40 weeks, respectively. Scans were obtained from the publicly available *dHCP* neonatal data set. One can see changes in visual texture rapidly taking place throughout brain development, e.g. as the brain matures, occurrence of dark intensities present in white matter regions gradually increases.

Studies in the neuroimaging literature suggest that changes in brain texture that take place during development are actually quantifiable. For instance, the developmental model presented by Towes et al. (Toews et al., 2012) hypothesized the existence of distinctive anatomical properties that can be localized in space and time, and can be used to represent structural development in neonatal MRI; conventional scale-invariant features proposed in (Lindeberg, 1998) and (Lowe, 2004) were used to successfully achieve this modeling. In fact, 3D scale-invariant features were recently shown to be incredibly powerful in capturing a keypoint signature 'brainprint' that could identify similarities in scans corresponding to unique adult subjects despite ageing and neurodegenerative disease progression (Chauvin et al., 2020), suggesting that both variations and consistencies in brain texture are quantifiable.

In addition to modeling the healthy brain, quantifying image textural patterns has shown success in the characterisation of pathology, as reported in a number of studies in paediatric neuro-oncology (Gutierrez et al., 2014; Orphanidou-Vlachou et al., 2014; Fetit et al., 2015). It is therefore important to establish a clear understanding of the role of visual texture when training deep learning models on neuroimaging data of the developing brain.

Our analysis empirically shows that deep CNNs can be trained to a high level of accuracy on complex neuroimaging segmentation tasks without being exposed to the underlying imaging data, but rather to explicit representations of the image's texture. We achieve this by encoding the imaging data as visual textural maps using the computationally simple *Local Binary Patterns (LBP)* algorithm (Ojala et al., 2002). We then train, validate, and test the networks directly on the resulting encoded maps, in a tissue segmentation problem. To this end, we use publicly available T2-weighted MRI scans of the developing neonatal brain obtained retrospectively from the *Developing Human Connectome Project* [1] *(dHCP)* cohort (Makropoulos et al., 2018; Bastiani et al., 2019).

## 2. Contribution and overview

This study is not the first to incorporate aspects of LBP with deep networks; work in the remote sensing literature looked into using a two-stream strategy for designing a CNN, where texture-encoded images were used as an additional stream that is fused with a standard RGB image pathway (Anwer et al., 2018). Work in computer vision illustrated that CNNs could be trained directly on LBP textural maps to achieve high levels of accuracy on face recognition tasks (Zhang et al., 2017). State-of-the-art work proposed the notion of local binary pattern *networks*, which uses binary operations as opposed to convolutions, and illustrated its utility on optical character recognition tasks (Lin et al., 2019).

To the best of our knowledge, however, our analysis is the first to demonstrate that CNNs could be used directly on explicit LBP textural maps in the complex task of image segmentation of developing human brain tissues, and on neuroimaging datasets in general. By developing accurate tissue segmentation models on explicit textural maps, the work offers two contributions to the fields of neuroimaging and deep learning:

- Firstly, it takes a step towards understanding the role of visual texture when training deep segmentation CNNs on heterogeneous neuroimaging data of the developing brain; an important area of research that has not been previously explored.

- Secondly, it contributes to the understanding of the inner workings of CNNs by showing empirical results that support the texture hypothesis, in line with recent findings on natural images. Evaluating these results suggests that CNNs do not necessarily require conventional images as input, and they may simply be capable of learning representations from well-structured spatial information.

In this regard, it is important to stress that our focus is not necessarily on improving segmentation accuracy using texture-encoding in this particular study, but rather to show that it is possible to achieve good performance using only texture-encoded maps as input to the CNNs for this complex neuroimage segmentation task.

---

1. developingconnectome.org

## 3. Materials and Methods

### 3.1. Image acquisition and pre-processing

558 three-dimensional, T2-weighted MRI scans were obtained retrospectively from the publicly available *Developing Human Connectome Project (dHCP)* neonatal cohort. Acquisition was carried out using a 3T Philips scanner and following a protocol described in (Makropoulos et al., 2018). Data was available in NIfTI format. Normalization of gray-level intensities was carried out by ensuring zero-mean and unit-variance within each scan. The scans have associated tissue labels that were generated using an automated segmentation pipeline. The pipeline was specifically designed for neonatal brain MRI using the well-established *Draw-EM* (Makropoulos et al., 2014) framework and was discussed in (Makropoulos et al., 2018). The labels were used as ground truth annotations and can be summarized as follows: 1. background (zero-intensity pixels), 2. cerebrospinal fluid (CSF), 3. cortical gray matter (cGM), 4. white matter (WM), 5. background bordering brain tissues, 6. ventricles, 7. cerebellum, 8. deep gray matter (dGM), 9. brainstem, and 10. hippocampus.

### 3.2. Structuring the dataset

Model-development set: 470 neonatal scans were included with the purpose of developing and optimizing models capable of segmenting developing brain tissues. Of the 470 scans, 450 were assigned for model training and 20 were used for validation throughout the training cycles. Subjects' postmenstrual age range was 24.7-42.1 weeks for the training data, and 27.6-42.2 weeks for the validation data. Held-out test set: 88 additional scans were completely held out from the model-development set; postmenstrual range was 24.3-42 weeks.

### 3.3. Generating LBP texture maps

In essence, the *LBP* algorithm assumes that the visual texture of an image can be characterized using two complementary measures: local spatial patterns and gray-level contrast (Pietikäinen, 2010). *LBP* is intensity invariant and computationally simple. It first considers the neighborhood of a given pixel of interest; variations in pixel intensities and positions then generate a systematic code that summarizes the local texture within the given pixel's neighborhood (Ojala et al., 2002). By computing a pixel-wise LBP code across the image a local-texture map can be produced, where every unit of the map is a representation of the spatial variation of intensities in the corresponding pixel's local neighborhood on the original image. The LBP code for a given pixel of interest can be formulated as:

$$LBP(x_c, y_c) = \sum_{p=0}^{P-1} f(i_p - i_c)2^p, \tag{1}$$

where $P$ is the number of sampling points, $i_c$ is the gray-level intensity of the pixel of interest defined by coordinates $(x_p, y_c)$, and $i_p$ is the gray-level intensity of the $p$th surrounding pixel. The binary pattern $f(x)$ is straightforward to compute:

$$f(x) = \left\{ \begin{array}{ccc} 1 & if & x \geq 0 \\ 0 & , & otherwise \end{array} \right. . \tag{2}$$

We applied uniform LBP operators to the original (pre-normalization) MRI scans using a 3x3 pixel neighborhood offset[2]. Two versions of the maps were computed using radius values of 1 and 10 pixels, respectively. *Scikit-image*'s local_binary_pattern module (Van der Walt et al., 2014) was used to carry out the computations. The output of the algorithm is an LBP map that has the same dimensions as the input image; see Figure 2.

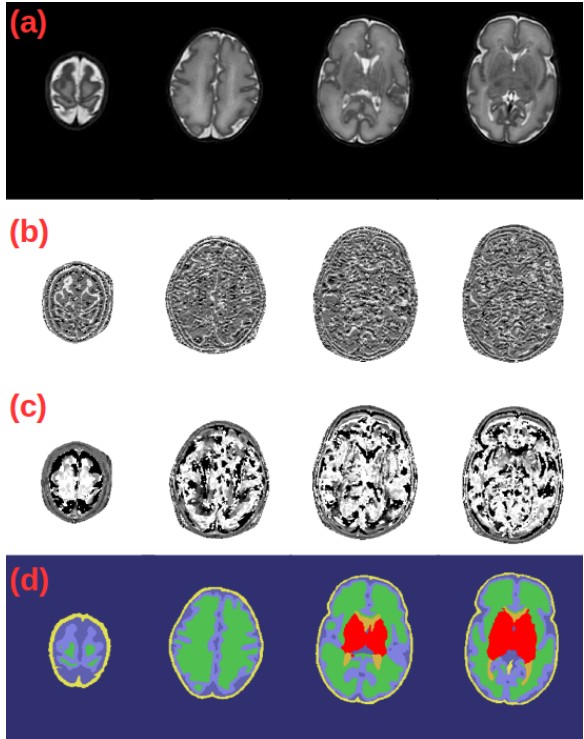

Figure 2: (a) Example axial slices from a T2-weighted scan in the *dHCP* neonatal cohort, (b) LBP map computed using a 1-pixel radius, (c) LBP maps computed using a 10-pixel radius, and (d) 10-class tissue labels generated by the *dHCP* structural pipeline and used as ground-truth for training segmentation models.

### 3.4. Training deep segmentation CNNs

We used the open-source *DeepMedic* framework (v0.7.0) (Kamnitsas et al., 2017) to train 10-class tissue segmentation networks using the tissue labels generated by the *dHCP* structural pipeline as ground truth. *DeepMedic* uses multiple complementary pathways; the primary one encodes local features, whereas the additional ones capture higher-level contextual information (Kamnitsas et al., 2017). In our set-up, the primary pathway had 8 layers, and each layer used a kernel dimension of 3x3x3. Residual connections were also used between the following layer pairs: 4 and 3, 5 and 6, 7 and 8. Two parallel sub-sampling pathways were used (8-layers deep); sub-sampling factors of [3, 3, 3] and [5, 5, 5] were applied along

---

2. Code is publicly available on Github.

the [x, y, z] axes, respectively. Each of the training cycles comprised 100 epochs, each consisting of 20 sub-epochs. In every sub-epoch, images were loaded from 5 cases, and 1000 segments were extracted in total. Training batch size was set to 5. Initial learning rate was set to 0.001 and was halved at predefined points using a scheduler (epochs 17, 22, 27, 32, 37, 42, 47, 52). Changing the mean and standard deviation of training samples was carried out in order to augment the dataset. Training was accelerated using an NVIDIA Tesla K80 graphics processing unit (GPU). In order to quantify segmentation performance, Dice similarity coefficient (DSC) was computed. When training directly on gray level intensities, the normalized MRI scans were used.

## 4. Results and Discussion

### 4.1. Results on validation data

When applied on the 20 full scans in the validation set, the model trained directly on gray level intensities achieved the following per-class DSC values across all subjects:

DSC: [ 0.9903, 0.8954, 0.9207, 0.9356, 0.8765, 0.7719, 0.8893, 0.9243, 0.9058, 0.7470 ].

This made use of labels generated by the well-validated *dHCP* structural pipeline in place of ground truth annotations. The DSC values corresponded to the following 10 classes:

Classes: [ zero-intensity background, CSF, cGM, WM, background bordering brain tissues, ventricles, cerebellum, dGM, brainstem, hippocampus ].

Evaluating the validation DSC values computed from output of the CNN trained on 10-pixel radius LBP maps showed that it achieved comparably high segmentation performance on the texture-encoded version of the data, specifically for the zero-intensity background, CSF, cGM, WM, ventricles, cerebellum, and dGM classes; the DSC values were only 1%-4% lower than those achieved with the T2-weighted scans. Performance on the background bordering brain tissues, brainstem and hippocampus tissues was also good, but lower by 5% - 11% than that achieved using the T2-weighted scans:

DSC: [ 0.9841, 0.8776, 0.8804, 0.9160, 0.8285, 0.7443, 0.8643, 0.8873, 0.8476, 0.6374 ].

*Vis-à-vis* the CNN trained on 1-pixel radius maps, the network still maintained high DSC values on validation data, albeit showing a substantial drop for the brainstem and hippocampus tissue classes:

DSC: [ 0.9781, 0.8573, 0.8865, 0.9128, 0.7867, 0.6680, 0.7034, 0.7692, 0.5408, 0.0026 ]. Note that for all three CNNs, inference was carried out on corresponding version of the data, e.g. the network trained on 1-pixel radius maps was validated on the 1-pixel radius map versions of the validation-set.

### 4.2. Results on held-out test data

We then evaluated the performance of the three models on 88 volumes in a completely held-out test set (example, see Figure 3). The results showed that all three CNNs achieved high DSC values on data completely unseen by the models before test time, with the exception of the brainstem and hippocampus tissue classes when 1-pixel LBP map CNN was used:

Using gray level intensities:
DSC: [ 0.9919, 0.9196, 0.9376, 0.9525, 0.8921, 0.8043, 0.9319, 0.9357, 0.9183, 0.7804 ].
Time for testing process: 11,193 seconds.

Using 10-pixel radius LBP maps:
DSC: [ 0.9869, 0.8825, 0.8949, 0.9221, 0.8458, 0.7610, 0.8954, 0.8926, 0.8377, 0.6505 ].
Time for testing process: 10,807 seconds.

Using 1-pixel radius LBP maps:
DSC: [ 0.9823, 0.8688, 0.9038, 0.9232, 0.8104, 0.6692, 0.7435, 0.7894, 0.5319, 0.0019 ].
Time for testing process: 11,101 seconds.

### 4.3. Discussion

Visual inspection of the results suggested that the CNN trained on 10-pixel radius LBP maps resulted in relatively more smooth segmentation maps compared to the one trained with 1-pixel maps; this indicates an element of trade-off between the complexity of the computed LBPs and how refined the segmentation output is. Additionally, the drop in performance on the hippocampus and brainstem classes when using 1 pixel radius as opposed to 10 suggests that the choice of granularity of the textural maps also has a direct effect on capturing changes in intensity within the classes of interest. Nevertheless, the performance of all three networks appeared invariant to PMA at scan, despite differences in texture and shape patterns across scans that belong to different ages.

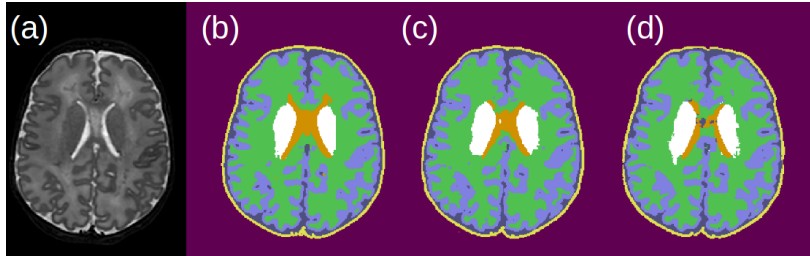

Figure 3: Example segmentation performance on (a) T2-weighted axial slice, using CNNs trained with (b) gray level intensities, (b) LBP maps computed using a 10-pixel radius, and (d) LBP maps computed using a 1-pixel radius.

To the best of our knowledge, the findings are the first to show that deep CNNs could be trained directly on textural maps to achieve successful segmentation performance on the

highly heterogeneous neonatal brain MRI data; these findings are also the first to show this on neuroimaging datasets in general. From a neuroimaging perspective, this is a crucial first step towards understanding the role of visual texture when training deep segmentation models on datasets of the developing brain. From a deep learning perspective, the findings question common consensus that CNNs perform well in computational perception tasks by learning complex shape hierarchies, and suggest a more important role to texture, at least in semantic segmentation tasks. Additionally, and since each unit on an LBP map is a representation of the textural neighborhood for the corresponding pixel in the original image, achieving segmentation success on LBP maps suggests that CNNs could simply be capable of learning representations from structured information, and do not necessarily require conventional images as input; an interesting area to explore in future work.

In terms of future work, it will also be interesting to explore whether the findings can generalize to other measures of texture, or whether they are specific to the LBP algorithm. Additionally, exploring the use of shape filters will be a natural next step. Further, it will be interesting to vary the complexity of the segmentation task by experimenting with more detailed segmentation maps that are also publicly available from *dHCP*, as exploring the relationship between the complexity of the segmentation task and the radius of the textural neighbourhood could give further interesting insights.

Having explicit textural and shape inputs to a CNN directly links to an active area of research referred to as representation disentanglement; please refer to the work by van Steenkiste for an empirical study on the topic (van Steenkiste et al., 2019), albeit not on textures. If disentangling the type of representations learned by complex networks could be carried out easily, this may have a direct impact on model performance when the training data is perturbed or when the application domain is shifted, potentially resulting in improved robustness against changes in gray level intensities, image acquisition protocols, or scanner hardware. Exploring this application within the context of developing brain MRI will drive our further future efforts.

## 5. Conclusion

We illustrated the feasibility of training deep segmentation CNNs on texture encoded input, using the computationally simple LBP algorithm, on heterogeneous MRI scans of the developing neonatal brain.

## Acknowledgments

The research leading to these results has received funding from the European Research Council under the European Union's Seventh Framework Programme (FP/2007-2013)/ERC Grant Agreement no. 319456. We are grateful to the families who generously supported this trial. This research was also supported by the UK Research and Innovation London Medical Imaging and Artificial Intelligence Centre for Value Based Healthcare.

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
