# OpenReview forum: "Training deep segmentation networks on texture-encoded input: application to neuroimaging of the developing neonatal brain"
_MIDL.io/2020/Conference — MIDL 2020_

### Official Review · AnonReviewer1 · 2020-03-13
**interesting work, failure to address major challenges and cite related texture work in neonatal brain MRI**

**Rating:** 2
**Confidence:** 5

**Summary:**

this paper proposes modeling neonatal MRI brain images in terms of binary texture operators. classification results are shown.


____________---_____--__-__--------_-_--_____________---_____--__-__--------_-_--_____________---_____--__-__--------_-_--_____________---_____--__-__--------_-_--_____________---_____--__-__--------_-_--_____________---_____--__-__--------_-_--_

**Strengths:**

front end texture operators in a deep learning context are interesting

____________---_____--__-__--------_-_--_____________---_____--__-__--------_-_--_____________---_____--__-__--------_-_--_____________---_____--__-__--------_-_--_____________---_____--__-__--------_-_--_


**Weaknesses:**

unfortunately, the authors are not aware

For example, texture operators have been used to predict , in the form of local 3D SIFT features
    to predict infant

'A feature-based developmental model of the infant brain in structural MRI' Toews et al., MICCAI 2013

Furthermore, the work of Toews et. al. performs age prediction in early stages of infant neuroevelopment, including the mylenation phase with contrast inversion between white and grey matter, from 0-104 weeks (0-2 years of age)

  It is unclear how the present approach would cope with appearances changes such as contrast inversion, since the work here is restricted to gestational ages at scan: 24.3 - 42.2 week, a rather narrow interval.

The primary challenge of Mylenation and contrast inversion typically occurs around 12 weeks, much early that the narrow 24.3 - 42.2 week investigated here. The authors appear unaware of developmental MRI changes, including mylenation.

The same 3D HoG-SIFT keypoint approach used to discover labelling errors in the OASIS, ADNI and HCP datasets previously unknown to the neuroimaging community, and source code is available, so the authors here should at least be aware they could potential compare to a highly effective texture-based approach:

Chauvin et al., NeuroImage 2020 'Neuroimage signature from salient keypoints is highly specific to individuals and shared by close relatives.'


**Detailed Comments:**

For example, the following statement is inaccurate:

   ' Importantly, changes in visual characteristics tend to occur rapidly during brain development as a result of the
continuous decrease of water content within the brain and the process of myelination (Serag, 2013); Figure 1. I '

Figure 1 shows the earliest neonatal MRI at 32 weeks, when mylenation is all but finished, no contrast inversion is observable.

A quick look at the following work (also uncited) would show that the primary challenges and rapid changes in infant MRI brain modeling occur in the first weeks of life, challenges which have been discussed and tackled at least several years ago.

VS Fonov, AC Evans, RC McKinstry, CR Almli and DL Collins Unbiased nonlinear average age-appropriate brain templates from birth to adulthood NeuroImage, Volume 47, Supplement 1, July 2009, Page S102 Organization for Human Brain Mapping 2009 Annual Meeting

**Justification Of Rating:**

Uncited literature, practical challenges, previous work.


____________---_____--__-__--------_-_--_____________---_____--__-__--------_-_--_____________---_____--__-__--------_-_--_____________---_____--__-__--------_-_--_____________---_____--__-__--------_-_--_____________---_____--__-__--------_-_--_____________---_____--__-__--------_-_--_

**Paper Type:**

both

**Questions To Address In The Rebuttal:**

Please correct discussion of challenges neonatal MRI modeling (e.g. age of mylenation),
   and cite and compare with existing work.



____________---_____--__-__--------_-_--_____________---_____--__-__--------_-_--_____________---_____--__-__--------_-_--_____________---_____--__-__--------_-_--_____________---_____--__-__--------_-_--_____________---_____--__-__--------_-_--_

**Special Issue:**

yes

---

> ### Author Response · Authors · 2020-03-26
> **The authors are sorry that an unfortunate misunderstanding on how they report the age at scan may have led to an incorrect assumption on how comprehensive the analysed dataset was**
>
> We thank the reviewer for taking the time to evaluate our paper. The reviewer has indeed made a number of good observations about interesting work in the literature which we will now incorporate into our paper. However, we would also like to note that there is an unfortunate misunderstanding on how we report the neonates’ ages at scan, particularly in the raised comments that relate to myelination, perhaps due to a lack of clarification from our part. We are grateful for all the suggestions made and have included our replies to specific comments below.
>
> The reviewer is right to point out that in previous research studies textural operators were indeed used to model the developing neonatal brain and predict outcomes of interest. The paper by Towes et al. is a good example as it makes use of well-established textural operators (namely SIFT) and illustrates that neonatal age prediction can be performed using such modelling. However, we would like to note a subtle point. Our study is focused on the role of textural representations within the context of deep learning and how this fits within neuroimaging of the neonatal brain, as opposed to the general topic of modelling the neonatal brain with textural attributes. Saying this, we agree with the reviewer that our paper would benefit from including such literature as this provides a broader context to the reader. We will now cite this paper and include a discussion on how conventional techniques differ from our approach, which involves training a deep network directly on textural representations.
>
> Also on the point of uncited literature, we support the reviewer’s assertion that recent work by Chauvin et al. has helped discover labelling inconsistencies on neuroimaging datasets such as ADNI and HCP with the use of HoG SIFT textural descriptors. Such work, as well as other work of similar nature in the literature, was initially not discussed in our paper due our focus on deep learning settings, as per the previous paragraph. In retrospect, we agree that discussing these findings is relevant and actually important in order to provide a more detailed context to the reader on the role of textural modelling in understanding the human brain. We will now update our paper accordingly.
>
> With regards to the reviewer’s good comments about myelination, e.g. “..the primary challenge of Mylenation and contrast inversion typically occurs around 12 weeks, much early that the narrow 24.3 - 42.2 week investigated here..”, we would like to bring the reviewer’s attention to an important point: the scale we use to report a neonate’s age relates to pregnancy and is measured from a relative starting point that is the mother’s last menstrual period before pregnancy. This is different from the chronological scale assumed by the reviewer, which is the number of weeks since the neonate’s birth. We have used the term ‘gestational age’ in our paper to refer to our scale, which in retrospect could be seen as inaccurate since we are referring to neonates as opposed to fetuses. We will now correct this to use the more accurate term, postmenstrual age (PMA).
>
> Having now clarified the issue of age scales, we hope the reviewer agrees that the large DHCP dataset we analysed was indeed acquired from neonates in their first few weeks of life, and is indeed representative of the primary challenges faced when analysing data from the developing neonatal brain as discussed in the work by Fonov et al. With regards to the reviewer’s good observation on Figure 1, we will now clarify and stress in the paper text that the scale we used is PMA, i.e. the examples in the figure are indeed representative of how the brain’s appearance during the very first few weeks of life.
>
> Finally, we would like to thank the reviewer again for the detailed review. We hope that our comments have now clarified the reviewer’s good points about myelination, and that the changes we will make alleviate the reviewer’s concerns about the related work in the literature. It is unfortunate that the misunderstanding about how we report the neonates’ ages might have given an incorrect impression that the work doesn’t address key challenges in neonatal imaging. We therefore invite the reviewer to kindly revisit the given rating and we would be happy to discuss any other points further.

---

> > ### Comment · AnonReviewer1 · 2020-04-04
> > **I would vote for accept**
> >
> > With the authors addressing similar work on the literature, this paper is acceptable.
> >
> > Thanks for clarifying the age, however in my understanding, there is little myelination 24.3 - 42.2 w gestational age here, starts something around 52 weeks (3 months following birth). This would be an additional challenge.

---

### Official Review · AnonReviewer4 · 2020-03-14
**Validation that CNNs Learn From Textures**

**Rating:** 2
**Confidence:** 4

**Summary:**

In this work, the authors attempt to provide additional evidence for the theory that CNNs rely heavily on texture information and mostly ignore shape information in the input. To support this theory they authors compute local binary patterns (LBP) of the input images then train and test CNNs based on these images for the task of segmentation from T2-weighted MRI brain scans. The authors show that performance is mostly maintained after using the LBP as compared to the pixel intensity values, with regions bordering the background class showing the most significant performance decrease.

**Strengths:**

1. The related works and motivation is clearly laid out. There have been several works in recent years (cited by the authors) which show CNNs are heavily biased towards texture information over shape information.
2. To the best of the reviewer's knowledge, this is the first work to focus on segmentation while examining the importance of texture over shape. An interesting analysis on the surface since, as opposed to classification and regression, segmentation requires fine-localization.

**Weaknesses:**

1. The contributions of this work are somewhat minimal. As the authors state, there have already been several works that show CNNs rely almost exclusively on texture information over shape, in standard computer vision data and even brain MRI.
2. While the authors try to go for a nice angle of demonstrating CNNs relying on texture for the task of segmentation, the algorithm works based on local binary patterns which remain highly-structured. Thus, to make the argument that shape information is not being learned is very hard to justify. When looking at Figure 2, one can see majority of the shape information is retained (and really shape information has to be retained for any segmentation network to have any chance of defining region boundaries).
3. There is no practical benefit to training and testing on LBP data that is apparent for this application domain.

**Justification Of Rating:**

This paper is clearly meant as a theory/validation paper. There are no practical benefits to the proposed method and no application benefits. However, from a theory point of view, the paper is just providing some minimal evidence of a theory which has already been shown in many works. The reviewer is not convinced that using local binary patterns for training can guarantee the network is using only texture information (i.e. all shape information has been removed). Further, there is nothing here which can provide some new key insights to future researchers and spawn new research areas.

**Paper Type:**

validation/application paper

**Questions To Address In The Rebuttal:**

1. How can the author justify that using LBP destroys all shape information (such that only texture is being learned).
2. Does examining the internals of the CNN trained on LBP given any key insights as compared to the one trained on intensities? Can any interesting patterns be discovered that can lead to new theories?

**Special Issue:**

no

---

> ### Author Response · Authors · 2020-03-26
> **Clarification on why the discussed approach relies on texture rather than shape, and why the work can lead to important theories**
>
> We thank the reviewer for the good critique of our work. We see the two questions raised by the reviewer as interesting and concise, and we include our answers and clarifications below.
>
> The first question raised by the reviewer relates to the concern that LBPs may not destroy all shape information, and that the networks may still rely on shapes for carrying out the segmentation tasks. We indeed cannot guarantee that LBP encoding completely destroys shape information, and we did not claim to do so in the paper. However, we know for sure that LBP is predominantly texture focused and we support our argument by referring to the original Texture Spectrum model proposed in 1990 (please see the work by He and Wang; “Texture Unit, Texture Spectrum, and Texture Analysis”), particularly the notion of “local” textures. Definitions of Texture Units and Texture Spectra directly relate to the LBP algorithm, and leads one to believe that LBP is a special case of the model.
>
> Hence, we are of the view that since LBP operates on local neighbourhoods, boundaries between different brain tissue classes might appear as though shape information is retained around them. We therefore argue that it may well be impossible to completely destroy shape information, since inter-class variation in local textures would always result in shape-like boundaries.
>
> Due to the way LBP works, any highly detailed image gets distilled into a simple set of integers that results in a map of a very low radiometric resolution. Such limited integers are capable of summarising spatial variations in pixel intensities (i.e. texture) but will certainly dilute morphometric features within an image (i.e. shape). Moreover, we believe that any retained shape features would not be directly impacted by variations in scale, hence, varying the length of the radius used to encode LBPs would always result in the same level of shape information. This is very different to, say, Canny edge detection, which predominantly keeps morphometric variations but ignores local texture. We are therefore confident that differences in performance observed across networks trained on different LBP maps are a direct result of changes in the encoded texture, as opposed to shape.
>
> In other words, algorithms can be more biased towards shape or texture. With regards to the reviewer’s good point that “shape information has to be retained for any segmentation network to have any chance of defining region boundaries”; this is exactly why we believe that our findings should be shared with the MIDL community, as they surprisingly show that predominantly texture-encoded input is sufficient for a complex segmentation task.
>
> The second good question raised relates to whether there is scope for further work to be carried out which can lead to interesting theories. The answer is “absolutely yes”. To start with, investigating whether findings from other texture or shape encoding techniques are consistent is a natural next step, as it may well be that CNNs can generally learn good representations from well-structured inputs. Secondly, the reviewer suggested examining the internals of the CNNs; we will do so by conducting a thorough investigation on how units get activated given different texture-encoded inputs, and how this can correlate to or differ from, say, shape-encoded input. This directly links to the new but important area on representation disentanglement, which can lead to models that learn explicit categories of texture or shape representations, or that attribute decisions to specific categories.
>
> This leads us to the third and perhaps most important direction, which is robustness in DL. If we can disentangle the type of representations learned by complex networks, this can have a direct impact on robustness when the training data is perturbed or when the application domain is shifted. For example, the MIDL community faces a key obstacle in translating DL into clinical practice, which is that models are sensitive to where training data is acquired. If disentangling, say, textural patterns from anatomy shapes is straightforward, models would learn to disregard textural changes below humans’ visual perception but are inherent to certain centres (e.g. artefacts of acquisition protocols, scanner manufacturer, post processing filters).
>
> In terms of applications to neonatal neuroimaging, we agree that our paper could benefit from highlighting the context of how visual texture can help understand the developing brain, as the readers may not be familiar with the broader literature. We invite the reviewer to refer to related comments by Reviewer 1 on the plethora of work on modelling the neonatal brain, as well as our reply.
>
> Finally, we thank the reviewer again for the good critique. We invite the reviewer to consider revisiting the recommendation, as we believe the MIDL community would benefit from our findings, particularly colleagues working on understanding the developing human brain.

---

> > ### Comment · AnonReviewer4 · 2020-04-02
> > **Updated Review**
> >
> > After further review of this work and the rebuttal made by the authors, the reviewer is inclined to support this work a little stronger. The authors arguments of LBP to essentially retain only texture information WITHIN brain regions is convincing, and thus segmentation can still be performed due to the differences between regions still allowing for the localization to take place. The contribution is still somewhat minimal considering the body of work in this area, but there is still a decent amount of novelty in this study and the reviewer does feel this should be shared with the MIDL community, and other researchers will benefit from seeing this work. Thank you to the authors for their detailed response to my comments. I look forward to seeing the stronger theoretical advancements coming from this study (representation disentanglement).
> >
> > UPDATED RATING: 3. Weak Accept.

---

### Official Review · AnonReviewer3 · 2020-03-14
**This is a nicely written and insightful submission with clear interpretation of the results.**

**Rating:** 4
**Confidence:** 4
**Recommendation:** Oral

**Summary:**

The premise is to test the texture hypothesis introduced in the ML literature, for the first time, in a newborn neuroimaging application. The goal is to represent neuroimaging data as local binary textural maps and learn accurate segmentation models using those instead of image-based representation. The image texture information is encoded in the form of local spatial patterns (LBP) that are easy to compute and are intensity invariant. DeepMedic is trained for a 10 class segmentation with data augmentation and the outcome is compared to the DHCP solutions via Dice overlap coefficients. The proposed pipeline is applied to an interesting and rich MRI data set from the recently released DHCP2 cohort of newborns.

**Strengths:**

The application area is new and well deserving. The results are promising when compared among intensity-based solution (segmentation from the default DHCP processing pipeline) and two different versions of the proposed pipeline.

**Weaknesses:**

I found the premise of the work interesting. I wish the authors took it a step further and looked at finer segmentation labels given that the Dice overlap metric is very forgiving when computed for large ROIs.  Using the more detailed segmentation (~90 labels) from the DHCP dataset could be more informative and could give more insight into design and interpretation. It might also help with the granularity vs ROI intensity profile discussion of the authors.


**Detailed Comments:**

Minor:
=======
Pawlowski, Glocker paper not hyperlinked
an 3x3 pixel --> a 3x3 pixel


**Justification Of Rating:**

It is a well-written paper, with well-thought-out experiments and insightful discussion of the results. The technology is not new, but the application area (pediatric neuroimaging) is, and the results are promising.

**Paper Type:**

validation/application paper

**Special Issue:**

yes

---

> ### Author Response · Authors · 2020-03-26
> **The authors are thankful for the positive feedback and will update the paper to include the suggestion on varying the complexity of the segmentation task in future work**
>
> We thank the reviewer for taking the time to review our paper and for appreciating our work, experimental methodology, and importance of findings in the neonatal neuroimaging domain.
>
> The suggestion on investigating whether our findings scale to a more granular segmentation problem of over 90 tissue classes is good and creative. We agree that this will be an excellent fit to the next steps of our work, particularly to investigate how varying the complexity of the segmentation task relates to choice of textural neighbourhood used to encode the images. It is plausible that as the number of tissue classes increases, using increasingly smaller radius values may be needed to maintain performance. We will now update our paper and add this point to the future work discussion.
>
> We also thank the reviewer for the observation on the broken hyperlink and typographical error. We will update our paper accordingly.
>
> Again, we appreciate the reviewer taking the time to evaluate our paper and for the positive feedback.

---

### Official Review · AnonReviewer2 · 2020-03-14
**A very preliminary study of using texture features instead of initial image intensity values, for classification on neuroimaging data**

**Rating:** 1
**Confidence:** 5

**Summary:**

This paper reports a study using texture features instead of initial image intensity values, for classification on neuroimaging data. Concretely, local binary pattern (LBP) is used for the experiments with two different radius values. A comparison with using initial image intensity values, however, does not show any advantages.

**Strengths:**

This work seems to be the first study of this kind on neuroimaging data. It is certainly something interesting to investigate for a particular application domain. The binary local pattern is also reasonable due to its popularity.

**Weaknesses:**

The technical novelty is low. In addition, several essential issues remain untouched. Overall, it is a very preliminary study only.

The finding is not a surprise. Basically, it confirms the results that have reported for other domains, in particular natural images. In all test cases the performance of using intensity values turns out to be, partly substantially, higher than using texture features. Therefore, what is the gain of this study? Given the extra information available in the texture information one may expect to receive higher performance using net architecture of the same complexity. Alternatively, one may expect to receive the same performance using a slimmer net architecture. Such important issues are not discussed.

There are other reasons why this is only an initial study only. Why are only radius values of 1 and 10 pixels used, not something in-between? The authors leave it future work to explore whether the findings can generalize to other measures of texture, or whether they are specific to the LBP algorithm. It is straightforward to repeat the same experiments for other popular texture features.


**Detailed Comments:**

It is better to put the results of the experimental results in table(s) to ease the comparison.

**Justification Of Rating:**

Despite of being the first study of this kind on neuroimaging data, this study is a very preliminary one only. A number of important issues remain untouched, or partly leaving as future work. The overall value of this study is unclear.

**Paper Type:**

validation/application paper

**Special Issue:**

no

---

> ### Author Response · Authors · 2020-03-26
> **Clarification on why the analysis offers important insights on the role of texture in CNNs, within the context of neonatal neuroimage segmentation**
>
> We thank the reviewer for the feedback on our work. The reviewer raises two main points upon which the given rating is based; we fully clarify both points below and invite the reviewer and AC to kindly revisit our paper.
>
> The first point raised relates to the “technical” novelty of the work.  We acknowledge the reviewer’s point in the sense that we did not claim to introduce a new algorithm or segmentation technique. However, we are in the view that good empirical work is needed and should be encouraged in the MIDL community. We also believe that the community should be particularly aware of our findings as they: (a) question fundamental understandings of how CNNs work, and (b) are indeed a first step towards understanding the role of visual texture when using CNNs on developing brain data. These, in addition to the neuroimaging domain novelty acknowledged by the reviewer, ought to be of tremendous interest to the community.
>
> On the same point of novelty and contributions, the reviewer mentions that “..the performance using intensity values turns out to be, partly substantially, higher than using texture features..”. Since common practice is to rely on point estimates (in this case, DSC) for evaluating performances of fairly complex set-ups, such statements can arguably only be supported with statistical hypothesis tests and robustness tests, which could confirm or reject that the differences of only a few percentages are indeed significant. We would like to stress that our motivation is to illustrate that fairly accurate segmentation can be achieved by training only on texture-encoded input without the network seeing the original data, and not necessarily to achieve a DSC that is a few percentages higher (the literature is already saturated with such work). We think, however, that our paper could benefit from stressing our motivation in the text, and we will do so to avoid any confusion.
>
> Further, and with all due respect to the reviewer, we felt that this comment is incorrect: “..The finding is not a surprise. Basically, it confirms the results that have reported for other domains, in particular natural images..”. As discussed thoroughly in our paper, the literature has reported conflicting findings on this topic. The shape hypothesis is actually the conventional understanding of how CNNs work, and is supported by various empirical findings (e.g. see Zeiler and Fergus, 2014; Ritter et al. 2017). Recent work, however, has suggested an important role to texture (e.g.  see Brendel and Bethge 2019), which motivated our novel study in the context of segmentation of complex structures in neuroimaging of the developing brain. We hope that this clarifies that the topic remains under debate in the literature and that exploring it within a neuroimaging context, particularly the complex area of understanding the developing brain, is of value to the community.
>
> We do not see why the reviewer has the impression that using only textural maps as opposed to T2-weighted data as an input to the networks has provided extra information, rather than diluted the information available. Where LBP maps were used, the networks were completely blinded to the original images, which is a key reason why we think our results are interesting, particularly for the complex problem of tissue segmentation in the developing brain. We invite the reviewer to cross-refer to Figure 2 for an illustration of how the input information is distilled to the networks when texture-encoding is carried out.
>
> We hope that the above discussion has addressed the reviewer’s first point on novelty and contributions. We believe these clarifications are sufficient reasons for revisiting our work.
>
> The second point raised in the review relates to the seemingly “preliminary” nature of our analysis. Whilst we agree that there are important next steps that ought to be carried out in this research, our study is certainly not preliminary. The study was carried out on 558 full volumes of neonatal brain MRI from a major connectomics project, a completely held-out test set was used for fair evaluation, a highly challenging 10-class tissue segmentation task was studied, representative granular data from a wide range of neonatal ages was included, and three separate sets of experiments were carried out. The suggestion about exploring slimmer architecture is also good and would indeed be interesting to explore from an optimisation perspective. However, we would like to reiterate that the goal of the paper is completely different and we did not claim to offer a new architecture/algorithm in this study.
>
> Finally, we would like to thank the reviewer for the time taken to go through our paper and for the feedback. We do not see the recommended rating as justifiable, partly since the reviewer acknowledges that the work is the first of its kind, but mainly because the goal of the paper is completely different from what seems to be assumed by the reviewer.

---

### Meta-Review · Area_Chair1 · 2020-04-07
**MetaReview of Paper230 by AreaChair1**

**Rating:** 3
**Recommendation For Accepted Papers:** Poster

**Metareview:**

Although methodological contributions are somewhat limited, the paper provides an interesting analysis on the role of texture for training a segmentation CNN, and demonstrates that texture in neonatal brain images can be used instead of original images to train the network. After the rebuttal, the majority of reviewers are in favour of accepting the paper.

**Paper Type:**

validation/application paper

**Special Issue:**

no

---

### Decision · Program_Chairs · 2020-04-11

Accept